# Root System Architecture Differences of Maize Cultivars Affect Yield and Nitrogen Accumulation in Southwest China

Song Guo [1], Zhigang Liu [2], Zijun Zhou [1], Tingqi Lu [3], Shanghong Chen [1], Mingjiang He [1], Xiangzhong Zeng [1], Kun Chen [1], Hua Yu [1], Yuxian Shangguan [1], Yujiao Dong [1], Fanjun Chen [2], Yonghong Liu [4] and Yusheng Qin [1,5,*]

1   Institute of Agricultural Resources and Environment, Sichuan Academy of Agricultural Sciences, Chengdu 610066, China; guosong1999@163.com (S.G.); zhouzijun1007@163.com (Z.Z.); 13908040304@163.com (S.C.); hemj9331@163.com (M.H.); xzhzeng@163.com (X.Z.); chenkun410@163.com (K.C.); yuhua353@163.com (H.Y.); 1987329002@sohu.com (Y.S.); echo-215@163.com (Y.D.)
2   College of Resources and Environmental Sciences, China Agricultural University, Beijing 100193, China; lzglzglhy@163.com (Z.L.); caucfj@cau.edu.cn (F.C.)
3   Mianyang Academy of Agricultural Sciences, Mianyang 621023, China; lutingqi0822@126.com
4   Crop Research Institute, Sichuan Academy of Agricultural Sciences, Chengdu 610066, China; 13908189593@163.com
5   Monitoring and Experimental Station of Plant Nutrition and Agro-Environment for Sloping Land in South Region, Ministry of Agriculture and Rural Affairs, Chengdu 610066, China
*   Correspondence: shengyuq@126.com; Tel.: +86-028-84504285

**Abstract:** Root system architecture (RSA) plays a critical role in the acquisition of water and mineral nutrients. In order to understand the root characteristics that contribute to enhanced crop yield and N accumulation high-yielding and N efficient cultivars under N-stressed conditions. Here, grain yield, N accumulation and RSA traits of six dominant maize cultivars (CD30, ZH311, ZHg505, CD189, QY9 and RY1210) grown in the Southwestern part of China were investigated in field experiment under three different N regimes in 2019–2020; N300 (300 kg N ha$^{-1}$), N150 (150 kg N ha$^{-1}$) and N0 (no N supplied). Using Root Estimator for Shovelomics Traits (REST) for the quantitative analysis of maize root image obtained in the field, RSA traits including total root length (RL), root surface area (RA), root angle opening (RO), and root maximal width (RMW) were quantified in this study. The results showed that Yield, N accumulation and RSA were significantly affected by N rates, cultivars and their interactions. Grain yield, N accumulation and root weight showed a similar trend under N300 and N150 conditions compared to N0 conditions. With the input of N fertilizer, the root length, surface area, and angle increase, but root width does not increase. Under the N300 and N150 condition, RL, RA, RO and RMW increased by 17.96%, 17.74%, 18.27%, 9.22%, and 20.39%, 18.58%, 19.92%, 16.79%, respectively, compared to N0 condition. CD30, ZH505 and RY1210 have similar RO and RMW, larger than other cultivars. However, ZH505 and RY1210 have 13.22% and 19.99% longer RL, and 11.41% and 5.17% larger RA than CD30. Additionally, the grain yield of ZH505 and RY1210 is 17.57% and 13.97% higher compared with CD30. The N accumulation of ZH505 and RY1210 also shows 4.55% and 9.60% higher than CD30. Correlation analysis shows that RL, RA, RO and RMW have a significant positive correlation with grain yield while RO and RMW have a significant positive correlation with N accumulation. Linear plus plateau model analysis revealed that when the RO reaches 99.53°, and the RMW reaches 15.18 cm, the N accumulation reaches its maximum value under 0–300 kg N ha$^{-1}$ conditions. Therefore, selecting maize cultivars with efficient RSA suitable for different soil N inputs can achieve higher grain yield and N use efficiency.

**Keywords:** maize; root system architecture; nitrogen rates; cultivars; yield

## 1. Introduction

The root is an essential organ in plants, and it plays an important role in nutrient uptake, growth and yield formation [1,2]. Root system architecture (RSA), including root

length, root numbers, root surface area, root angle and root width, is an important trait in crops for the acquisition of underground resources [3,4]. The RSA of maize is influenced by genotype and environmental factors such as water, nutrients and temperature [5,6]. Nitrogen (N) is the key limiting nutrient in crop production. At the same time, Ninputinagricultural systems are also an important factor affecting environmental degradation and climate change [7]. Enhancing N use efficiency (NUE) is one of the most effective ways in sustainable agriculture to meet the 2050 global food demand projected [7,8]. Understanding the relationship between N uptake and utilization efficiency and RSA in maize is an important step towards improving maize productivity. Breeding new varieties based on RSA differences will improve N use efficiency (NUE) in maize production [3,9].

Modern varieties with deeper root distribution can increase yield under low N conditions [10]. A strong root system is an important factor for high yields [11]. There is a positive correlation between root weight and above-ground biomass and ultimately yield [12]. Under high planting density, a medium root system with more root distribution is more likely to result in high yield [6]. The interaction between RSA and soil N absorption determines yield largely [6,13]. Lynch considered that steep, cheap and deep are ideal RSA for obtaining N fertilizer and water in a low N input system in maize [5]. The maize root length and surface of different eras showed an increasing trend followed by a decreasing trend in China [14]. It has been reported that the effect of root horizontal distribution on grain yield is greater than that of root horizontal distribution [15]. Under low N conditions, the root horizontal expansion decreased [16]. The RSA of modern maize varieties in China is characterized by "horizontal contraction and vertical extension", which is more suitable for planting at a higher N level [17]. Response to N fertilizer varies with genotype, the yield and root biomass of maize varieties with high N efficiency are higher than those with low N efficiency [10,18].

High temperature, drought, poor soil fertility and nutrient leaching are persistent agronomic challenges in spring maize production in the central Sichuan Basin [19]. Thus, it is crucial to identify maize varieties with an ideal RSA suitable for this environment. Root Estimator for Shovelomics Traits (REST) is a simple, rapid and effective method for the quantitative analysis of plant root images obtained in the field [20–22]. In this study, we used this high-throughput root phenotype analysis method to study the genotypic differences and N response of RSA among maize varieties grown in this area.

## 2. Material and Methods

### 2.1. Plant Materials

A field experiment was conducted at Sichuan Agricultural Research Institute Modern Agriculture Experimental Station, Deyang City, Sichuan Province (30°36.784′ N, 105°01.322′ E) in 2019–2020. A total of 6 maize varieties were tested (Table 1), all of which are currently dominant high-yield spring maize cultivars in the hilly region of central Sichuan, namely Chengdan 30 (CD30), Zhenghong505 (ZH505), Zhenghong 311 (ZH311), Chuandan 189 (CD189), Quanyu 9 (QY9) and Rongyu 1210 (RY1210).

**Table 1.** Characteristics of cultivars used in this study.

| Cultivar | Year of Release | Parents | Breeding Institution |
|---|---|---|---|
| CD30 | 2004 | Chengzi2142 × Zhengzi205-22 | Sichuan Academy of Agricultural Sciences |
| ZH311 | 2006 | K236 × 21-ES | Zhenghong Seeds Co., Ltd. (Chengdu, China) |
| ZH505 | 2008 | K305 × K389 | Zhenghong Seeds Co., Ltd. (Chengdu, China) |
| CD189 | 2009 | SCML203 × SCML1950 | Sichuan Agricultural University |
| QY9 | 2011 | Y3052 × 18-599 | Sichuan Academy of Agricultural Sciences |
| RY1210 | 2015 | SCML202 × LH8012 | Sichuan Agricultural University |

### 2.2. Experimental Design

The experimental design was a split-plot with three replicates, with N fertilizer treatments in the main plots and the cultivars in the subplots. The variable between plots was three application rates of N fertilizer, namely 300 kg N ha$^{-1}$ (N300), 150 kg N ha$^{-1}$ (N150), and no N supplied (N0). The plots were fertilized with 90 kg ha$^{-1}$ P$_2$O$_5$ and 150 kg ha$^{-1}$ K$_2$O. Phosphorus and potassium fertilizers were applied before sowing, 50% of the N fertilizer was applied as a base dressing, and the remaining 50% was applied at the stem jointing stage in N300 and N150 treatments. The subplots area was 20 m$^2$ (5-m-long × 4-m-width). Maize was over-seeded on 1 April in 2019 and 7 April in 2020. At the V3 stage, seedlings were thinned to a final density of 50,000 plants ha$^{-1}$, which is the optimum population for maize cultivars grown in local areas. Cultivars were harvested on 11 August in 2019 and 15 August in 2020. The field was irrigated before sowing. Plots were kept free of weeds, insects and diseases with chemicals based on standard practices. No irrigation was applied during the growing season. Rainfall throughout the growing season was as shown in Figure 1.

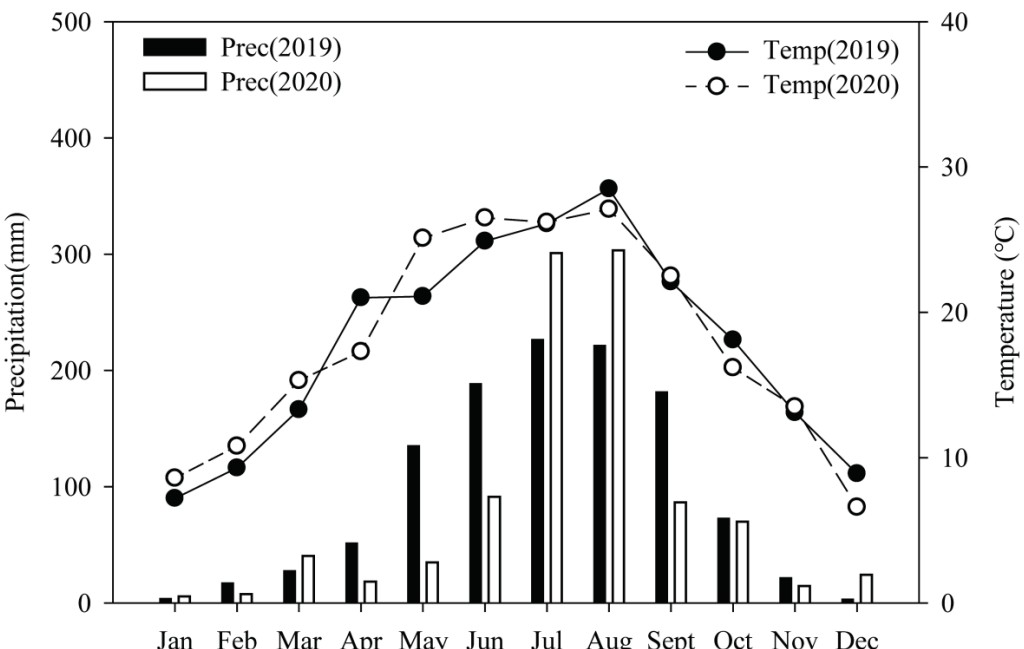

**Figure 1.** Temperature and precipitation of Zhongjiang County during the study period in 2019 and 2020. Note: Prec, Total monthly precipitation (mm); Temp, Average monthly temperature (°C).

Soil physical and chemical characteristics were evaluated at the beginning of the experiment for each treatment by analyzing three soil samples. The topsoil layer (0–30 cm) contained organic matter 11.5 g kg$^{-1}$, total N 0.99 g kg$^{-1}$, alkali-hydrolyzable N 74.0 mg kg$^{-1}$, available phosphorus (Olsen-P) 14.5 mg kg$^{-1}$, ammonium acetate extractable potassium (K) 172 mg kg$^{-1}$ and pH 7.94 (1:1.25 *g/v*) in N300 treatment. The chemical characteristics in N150 treatment were as follows: organic matter 10.8 g kg$^{-1}$, total N 0.81 g kg$^{-1}$, alkali-hydrolyzable N 32.5 mg kg$^{-1}$, Olsen-P 17.5 mg kg$^{-1}$, ammonium acetate extractable potassium (K) 161.8 mg kg$^{-1}$ and pH 7.98. In No treatment, chemical characteristics were organic matter 9.6 g kg$^{-1}$, total N 0.69 g kg$^{-1}$, alkali-hydrolyzable N 8.7 mg kg$^{-1}$, Olsen-P 11.9 mg kg$^{-1}$, ammonium acetate extractable potassium (K) 143.7 mg kg$^{-1}$ and pH 7.85. The soil type is brown and is classified as Cambisols with sandy loam according to the FAO classification system (IUSS Working Group WRB, 2015).

### 2.3. Agronomic Trait Measurements

At silking and physiological maturity stage, three uniform plants from each plot were cut at the soil surface and separated into leaves, stem and grain (only at maturity). At the silking stage, roots were excavated within a soil volume of 30 cm (length) × 30 cm (width) × 25 cm (depth) for each plant and were then shaken off a large fraction of the soil adhering to the root system. Afterward, the roots were washed under low pressure using a water hose and nozzle. Root imaging and processing were as described by Colombi et al. [21]. Briefly, root images were captured in the imaging tent using a digital camera (Canon EOS 400D, Canon, Tokyo, Japan) with a 35 mm fixed focal lens (Canon EF 35 mm f/2.0, Canon, Tokyo, Japan). The image size was 35 × 52.5 cm resulting in a pixel size of 0.13 mm. Root images analyses were processed using REST software (Figure 2). RSA traits, including total root length, surface area, angle opening, and maximal width, were quantified in this study.

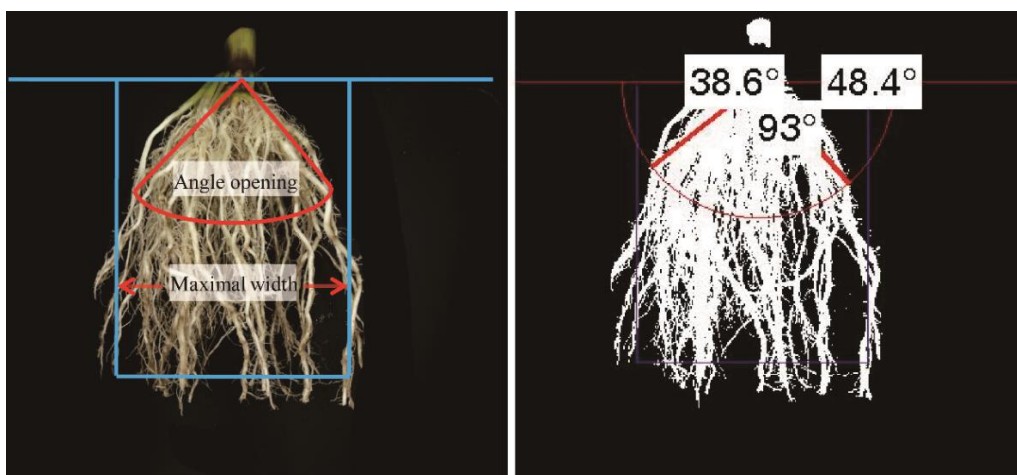

**Figure 2.** Root image processing with Root Estimator for Shovelomics Traits (REST).

At both sampling dates, all samples were heat-treated at 105 °C for 30 min, dried at 65 °C to constant weight. After obtaining dry matter weight, the samples were ground into fine powder for N measurement. N concentration at silking (leaves and stem) and maturity (leaves, stem, and grain) were determined by the semi-micro Kjeldahl procedure [23]. At maturity, the ears in each plot were harvested. Grain was oven-dry to determine grain moisture content. The grain yield was determined and then standardized to 14% moisture.

### 2.4. Statistical Analysis

Statistical analysis was performed using SPSS Statistics 17.0 (SPSS, Inc., Chicago, IL, USA). Three-way analysis of variance (ANOVA) was used to test for significant differences among N treatment, cultivar, year and N treatment × cultivar × year interaction. N level and cultivar were treated as a fixed effect. The least significant difference test (LSD) was used to evaluate significant differences between means when a significant effect was detected by ANOVA. Means for each cultivar were used for correlation analysis and linear platform model fitting.

## 3. Results

### 3.1. Grain Yield and N Accumulation Properties

Across the two years, significant differences in grain yield and N accumulation were found among N treatments (Tables 2 and 3). ANOVA showed significant effects of N levels (0, 150 and 300 kg ha$^{-1}$) (N), cultivar (C), and years (Y) on grain yield, ED, KPR and N accumulation at silking and maturity. The interaction of N × C had a significant effect on N accumulation at the silking stage, and grain yield and N accumulation at maturity. Due to the less rain from April to June 2020 (Figure 1), the grain yield and N accumulation were significantly lower than those in 2019.

The yield in N300 was similar to that of N150, with both of them having higher yields than N0, except that there was no difference in 2019. N accumulation showed a similar trend under N300 and N150 conditions compared to N0 conditions. The grain yield and N accumulation among cultivars were significantly different. The grain yield per plant ranged from 126.16 to 148.33 g, and the maximum value was 17.58% in ZH505, higher than the cultivar with the minimum value, CD30 (Table 2). There was no significant difference in grain yield betweenZH505, ZH311, CD189 and RY1210, although they were significantly higher than CD30. The grain weight of RY1210 between 2019 and 2020 had no difference at the N300 level. At the silking stage, the difference of N accumulation between varieties was mainly on the leaf. ZH311 showed the highest value in N accumulation, and 10.95% higher than CD30 which has the minimum value. However, the difference in N accumulation between varieties was mainly on leaf and grain at maturity. The minimum value of total N accumulation was observed in QY9 and 10.57% lower when compared with RY1210 (Table 3).

**Table 2.** Analysis of variance in GW, HKW, EL, ED, RPE and KPR on maize of six cultivars under three N conditions.

| Treatment | GW (g Plant$^{-1}$) | HKW (g) | EL (cm) | ED (cm) | RPE | KPR |
|---|---|---|---|---|---|---|
| Nitrogen (N) | | | | | | |
| N0 | 127.12 b | 28.52 b | 15.81 a | 47.72 b | 16.44 a | 31.847 b |
| N150 | 149.02 a | 32.57 a | 16.54 a | 49.65 a | 16.71 a | 35.667 a |
| N300 | 149.11 a | 32.17 a | 17.93 a | 48.26 ab | 16.35 a | 35.639 a |
| Cultivar (C) | | | | | | |
| CD30 | 126.16 c | 29.55 c | 16.08 a | 46.07 c | 17.08 b | 34.03 b |
| ZH311 | 146.03 ab | 33.61 a | 18.06 a | 50.37 a | 16.42 b | 31.31 c |
| ZH505 | 148.33 a | 26.21 d | 17.78 a | 49.84 a | 19.17 a | 34.64 ab |
| CD189 | 145.74 ab | 32.41 ab | 15.75 a | 48.02 b | 14.86 c | 36.06 a |
| QY9 | 140.46 b | 31.0 bc | 16.47 a | 49.39 a | 16.25 b | 35.83 ab |
| RY1210 | 143.79 ab | 33.74 a | 16.42 a | 47.59 b | 15.22 c | 34.44 ab |
| Year (Y) | | | | | | |
| 2019 | 157.72 a | 31.50 a | 17.43 a | 50.08 a | 16.89 a | 38.02 a |
| 2020 | 125.78 b | 30.68 a | 16.09 a | 47.01 b | 16.11 b | 30.75 b |
| Source of variation | | | | | | |
| N | ** | ** | ns | ** | ns | ** |
| C | ** | ** | ns | ** | ** | ** |
| Y | ** | ns | ns | ** | ** | ** |
| N × C | * | ns | ns | ** | ns | * |
| N × Y | ** | ns | ns | ** | ns | ns |
| C × Y | * | ns | ns | * | ns | * |
| N × C × Y | ns | ns | ns | * | ns | ns |

Within N or cultivar or year, numbers followed by different letters indicate significant difference ($p < 0.05$). * significant at $p < 0.05$, ** significant at $p < 0.01$, ns: not significant ($p > 0.05$). GW, Grain weight; HKW, hundred-kernel weight; EL, Ear length; BHL, Bald head length; ED, Ear diameter; RPE, Rows per ear; KPR, Kernels per row.

### 3.2. Root System Architecture Traits Evaluation

Variance analysis of root traits showed that N treatments had a significant effect on root traits (Table 4). The interaction of the N × cultivar had a significant effect on all root traits. Therefore, the N × cultivar was further analyzed. Root dry weight, total length, surface area, and angle opening in N150 were similar to that of N300, with both of them having higher values than the N0 condition. In addition, the maximal width in N150 was higher than that of N300 and N0 treatments. Under N150 condition, the root angle, root width, root length and root surface area increased by 19.93%, 16.79%, 20.39% and18.58% compared with no fertilizer treatment. The RW among cultivars was significantly different. The maximum value of RW was 20.36 g in RY1210 higher than the cultivar with the minimum value, QY9. The maximum value of total length was 1651.29 cm in RY1210 higher than

the cultivar with the minimum value, CD30. The maximum value of the surface area was 2851.67 cm$^2$ in ZH505 higher than the cultivar with the minimum value, CD189. The root angle opening of RY1210 and CD30 were larger than others. The root maximal width of ZH 505, RY1210 and CD30 were larger than others.

**Table 3.** Evaluation of N accumulation traits of six cultivars under three nitrogen (N) conditions.

| Treatment | N Accumulation at Silking (g plant$^{-1}$) | | | N Accumulation at Maturity (g plant$^{-1}$) | | | |
|---|---|---|---|---|---|---|---|
| | **Stem** | **Leaf** | **Total** | **Stem** | **Leaf** | **Grain** | **Total** |
| Nitrogen (N) | | | | | | | |
| N0 | 0.50 b | 0.57 b | 1.07 b | 0.37 b | 0.20 c | 0.87 b | 1.44 b |
| N150 | 0.66 a | 0.90 a | 1.56 a | 0.62 a | 0.34 b | 1.35 a | 2.31 a |
| N300 | 0.65 a | 0.92 a | 1.57 a | 0.59 a | 0.36 a | 1.38 a | 2.33 a |
| Cultivar (C) | | | | | | | |
| CD30 | 0.56 a | 0.76 bc | 1.32 b | 0.51 a | 0.34 a | 1.14 c | 1.98 b |
| ZH311 | 0.62 a | 0.84 a | 1.46 a | 0.54 a | 0.29 bc | 1.14 c | 1.97 b |
| ZH505 | 0.62 a | 0.78 bc | 1.39 ab | 0.51 a | 0.30 bc | 1.26 ab | 2.07 ab |
| CD189 | 0.62 a | 0.82 ab | 1.43 ab | 0.57 a | 0.31 ab | 1.15 c | 2.03 ab |
| QY9 | 0.64 a | 0.73 c | 1.37 ab | 0.49 a | 0.28 c | 1.17 bc | 1.94 b |
| RY1210 | 0.58 a | 0.84 a | 1.42 ab | 0.53 a | 0.28 c | 1.35 a | 2.17 a |
| Year (Y) | | | | | | | |
| 2019 | 0.62 a | 0.88 a | 1.50 a | 0.64 a | 0.35 a | 1.22 a | 2.21 a |
| 2020 | 0.59 a | 0.71 b | 1.30 b | 0.41 b | 0.25 b | 1.18 a | 1.84 b |
| Source of variation | | | | | | | |
| N | ns | ** | ** | ** | ** | ** | ** |
| C | ** | ** | * | ns | ** | ** | * |
| Y | ns | ** | ** | ** | ** | ns | ** |
| N × C | ns | ** | * | * | * | ns | * |
| N × Y | ns | ns | ns | ns | ** | ns | * |
| C × Y | ns | ** | * | ns | ns | ** | ns |
| N × C × Y | ns | * | ns | ns | ** | * | ns |

Within N or cultivar or year, numbers followed by different letters indicate significant difference ($p < 0.05$). * significant at $p < 0.05$, ** significant at $p < 0.01$, ns: not significant ($p > 0.05$).

**Table 4.** Evaluation of root system traits of six cultivars under three nitrogen (N) conditions.

| Treatment | Root Weight (g Plant$^{-1}$) | Total Length (cm) | Surface Area (cm$^2$) | Angle Opening (°) | Maximal Width (cm) |
|---|---|---|---|---|---|
| Nitrogen (N) | | | | | |
| N0 | 12.21 b | 1325.30 b | 228.56 b | 88.08 b | 15.08 c |
| N150 | 19.81 a | 1595.54 a | 271.03 a | 105.63 a | 17.61 a |
| N300 | 19.39 a | 1563.32 a | 269.11 a | 104.17 a | 16.47 b |
| Cultivar (C) | | | | | |
| CD30 | 17.30 b | 1376.23 c | 252.83 bc | 104.42 a | 17.63 a |
| ZH311 | 17.09 b | 1551.27 b | 255.06 bc | 97.19 b | 15.67 bc |
| ZH505 | 17.28 b | 1558.24 b | 281.67 a | 97.62 b | 17.08 a |
| CD189 | 16.56 b | 1353.67 c | 249.44 c | 97.40 b | 16.02 b |
| QY9 | 14.22 c | 1477.61 b | 232.50 d | 93.74 b | 14.79 c |
| RY1210 | 20.36 a | 1651.29 a | 265.89 b | 105.41 a | 17.11 a |
| Year (Y) | | | | | |
| 2019 | 22.76 a | 1863.40 a | 284.17 a | 101.05 a | 16.45 a |
| 2020 | 11.52 b | 1126.04 b | 228.30 b | 97.55 b | 16.32 a |
| Source of variation | | | | | |
| N | ** | ** | ** | ** | ** |
| C | ** | ** | ** | ** | ** |
| Y | ** | ** | ** | * | ns |
| N × C | ** | * | ** | * | ** |
| N × Y | ** | ns | ns | ns | ns |
| C × Y | * | ** | ns | ns | ns |
| N × C × Y | ** | * | ** | * | ** |

Within N or cultivar or year, numbers followed by different letters indicate significant difference ($p < 0.05$). * significant at $p < 0.05$, ** significant at $p < 0.01$, ns: not significant ($p > 0.05$).

The root dry weights under N150 and N300 were increased by 62.24% and 58.80%, respectively, compared with N0 (Figure 3). It indicated that N application might increase maize root weight, while excessive N will reduce it (Figure 4). The root dry weight was also significantly different among cultivars. Under N0 treatment, RY1210 showed higher root weights, while QY9 had the lowest values in two years. Under N150 treatment, RY1210, and CD189 had higher root weights, while QY9 had the lowest values. The root weight under N300 treatment, of which CD189 had the least root weight compared to other varieties. CD189 were very sensitive to N stress, and the root weight decreased significantly under N deficiency or excess.

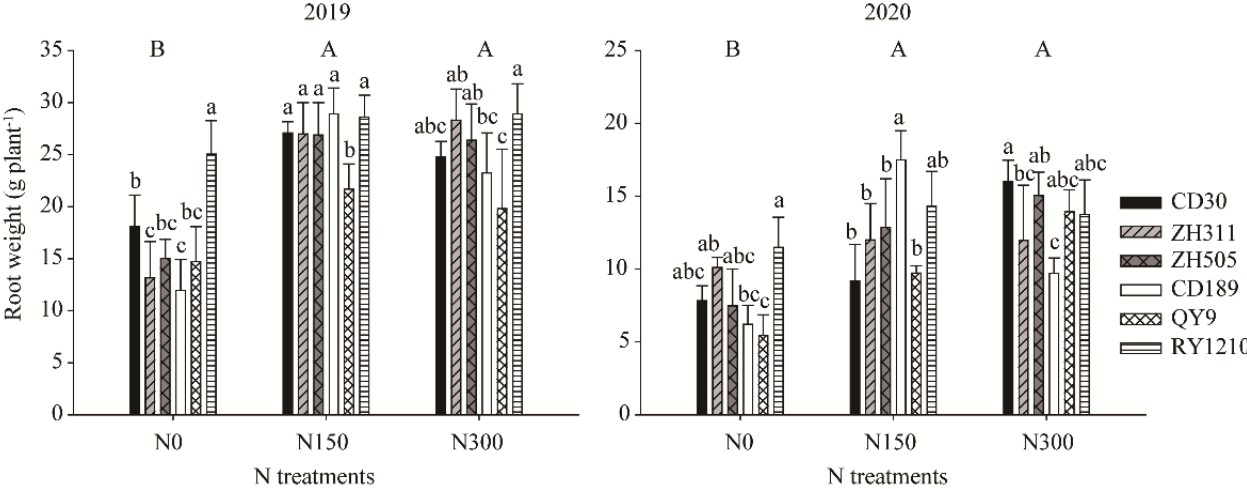

**Figure 3.** Root biomass of six maize cultivars under three N treatments in 2019 and 2020. Bars indicate standard error. Different lower case letters indicate significant differences at *p* < 0.05 among the cultivars in the same N treatment. Different capital letters indicate significant differences among the N treatments.

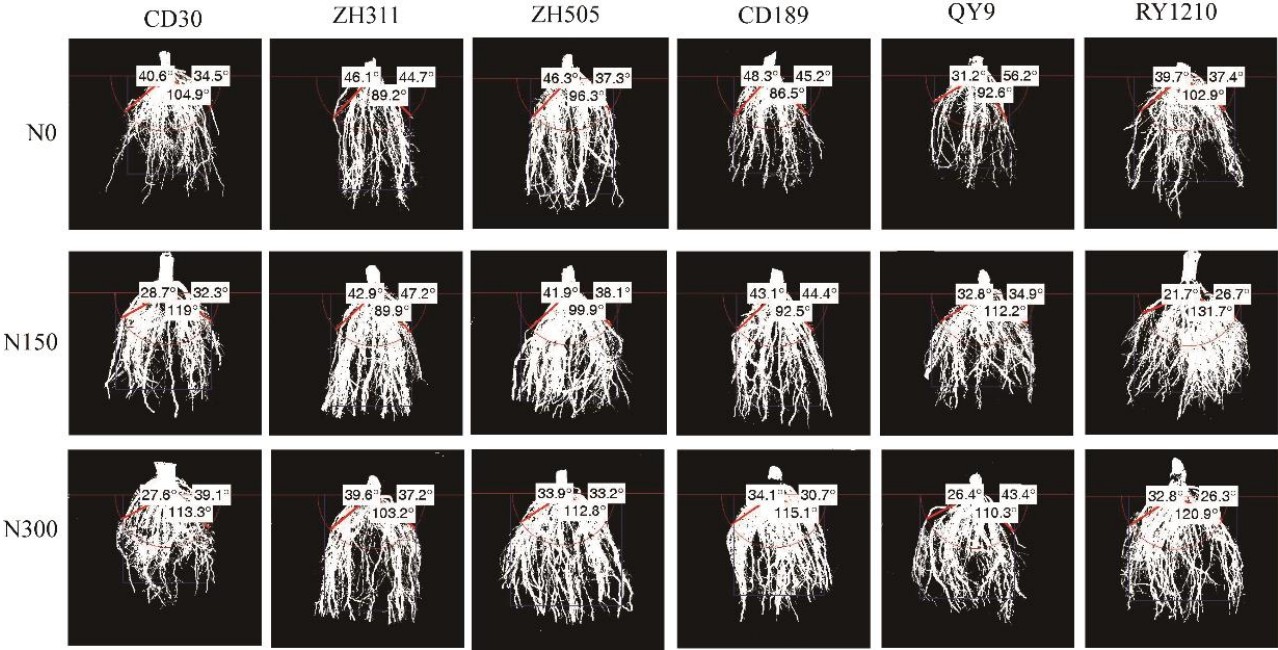

**Figure 4.** The distribution of the root system of six maize cultivars in three N treatments.

The total root length and root surface area were regulated by N fertilizer (Figure 5). There was no significant difference in root total length and surface area between N150 and

N300, which were higher than that of N0. Compared with N0, the root length and root surface area of N150 were increased by 20.39% and 18.58%, respectively. The root length and root surface area of N300 were increased by 17.96% and 17.74%, respectively (Table 4). This shows that N can promote the growth of roots. However, excessive application of N fertilizer inhibits root elongation and root surface area increase. The root length and root surface area of most maize varieties showed a trend of increasing at first and then decreasing at the three N levels, while RY1210 and CD189 showed an increasing trend all the time, indicating that RY1210 and CD189 were not sensitive to high N. The root length of RY1210 under N0 treatment is higher than that under N300 treatment. Root surface area under N0 is similar to that of N300. It indicates that this genotype is not sensitive to low N stress and belongs to low N tolerance varieties.

The root angle and width are regulated by N as well (Figure 5); they increased at first and then decreased with the increase in N supply, while the difference between N150 and N300 was significant in root width. The root angles of N150 and N300 were increased by 19.92% and 18.27% compared to the N0 treatment, respectively. The root maximal width of N150 and N300 was increased by 16.79% and 9.22% compared with the N0 treatment, respectively. These results indicate that N might promote the growth of maize roots, and the angle and width of the root system are significantly increased. However, excessive N may inhibit the increase in root angle and width.

The root angle opening and width between genotypes were significantly different. Under three N treatments, CD30 and RY1210 had larger root angles, CD189 and QY9 showed smaller ones, while ZH505 had a larger root width at N150 and N300, respectively. However, the change in root width was not completely consistent with the root angle. Under three N treatments, CD30, ZH505 and RY1210 had larger root widths, while QY9 had a smaller root width.

### 3.3. Relationship between RSA, Grain Yield, and N Accumulation

Significant correlations were found between RSA, grain yield and N accumulation (Figure 6). The grain yield increased logarithmically with increasing root length and surface area, and the regression multiple $R^2$ values were 0.84 and 0.69 ($p < 0.01$). Furthermore, N accumulation increased logarithmically with increasing root length and surface area ($p < 0.01$). With the increase of root length of maize roots, the grain yield and N accumulation continued to increase, while after reaching a certain point, the yield and N accumulation did not continue to increase, but showed a downward trend (Figure 7).

Yield and N accumulation are significantly related to root angle and root width (Figure 8). With the increase of root angle and root width, the yield and N accumulation continue to increase, while their continuous increase cannot further increase yield. On the contrary, too large root width will reduce maize yield and N accumulation. The trends in yield and N accumulation at the three N levels are consistent with the linear + plateau model ($p < 0.05$). Results from the model showed that when the root angle reaches 99.53° and 97.39°, the N uptake will reach the plateau value of 2.56 g plant$^{-1}$ and 2.11 g plant$^{-1}$ in 2019 and 2020, respectively; when the root width reaches 15.18 cm and 14.83 cm, the y N accumulation plateau value will be 2.34 g plant$^{-1}$ and 1.90 g plant$^{-1}$ in 2019 and 2020, respectively. Therefore, when the root angle of cultivars reaches 99.53°, and the root width reaches 15.18 cm, higher yield and N accumulation can be obtained.

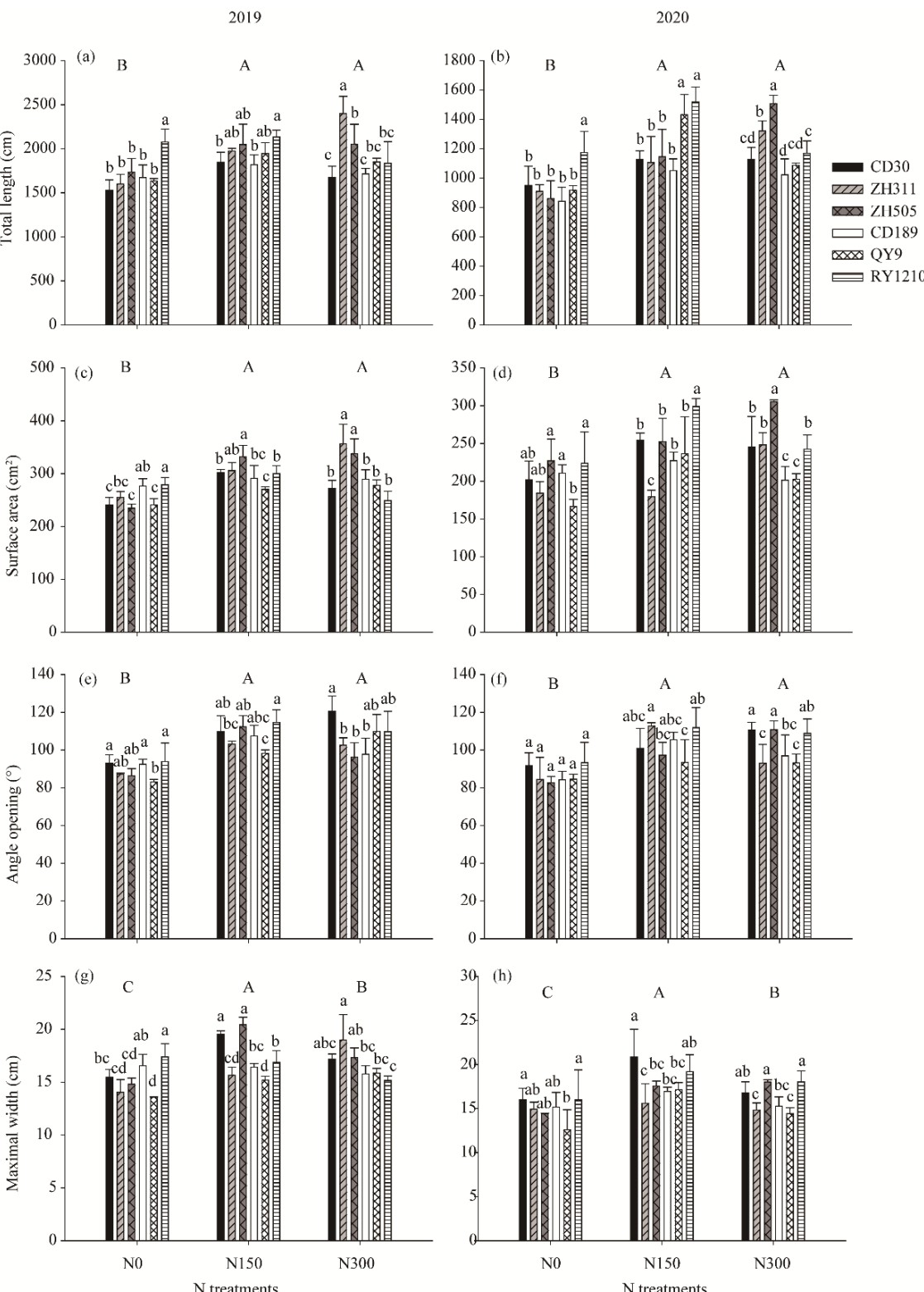

**Figure 5.** Root total length, surface area, angel opening, and maximal width of six maize cultivars under three N treatments in 2019 and 2020: (**a**) root total length in 2019, (**b**) root total length in 2020, (**c**) root surface area in 2019, (**d**) root surface area in 2020, (**e**) root angel opening in 2019, (**f**) root angel opening in 2020, (**g**) root maximal width in 2019, (**h**) root maximal width in 2020. Bars indicate standard error. Different lower case letters indicate significant differences at *p* < 0.05 among the cultivars in the same N treatment. Different capital letters indicate significant differences among the N treatments.

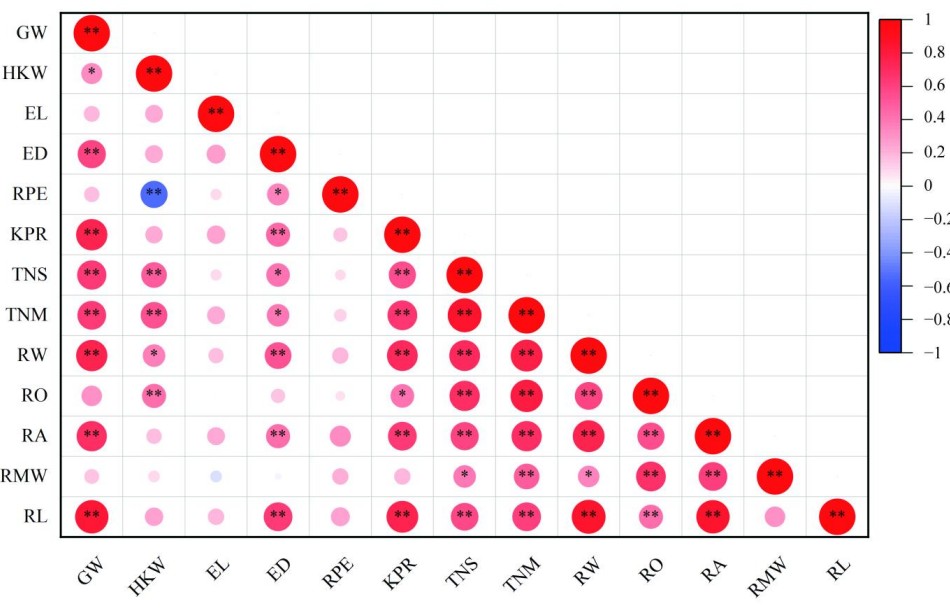

* p<=0.05  ** p<=0.01

**Figure 6.** Correlation coefficients between grain yield, N accumulation, and root system traits of six cultivars under three nitrogen (N) conditions.  * significant at the 0.05 probability level; ** significant at the 0.01 probability level.  GW, Grain weight; HKW, hundred-kernel weight; EL, Ear length; BHL, Bald head length; ED, Ear diameter; RPE, Rows per ear; KPR, Kernels per row; TNS, total N accumulation at silking; TNM, total N accumulation at maturity; RW, root weight; RO, root angle opening; RA, root surface area; RMW, root maximal width; RL, root length.

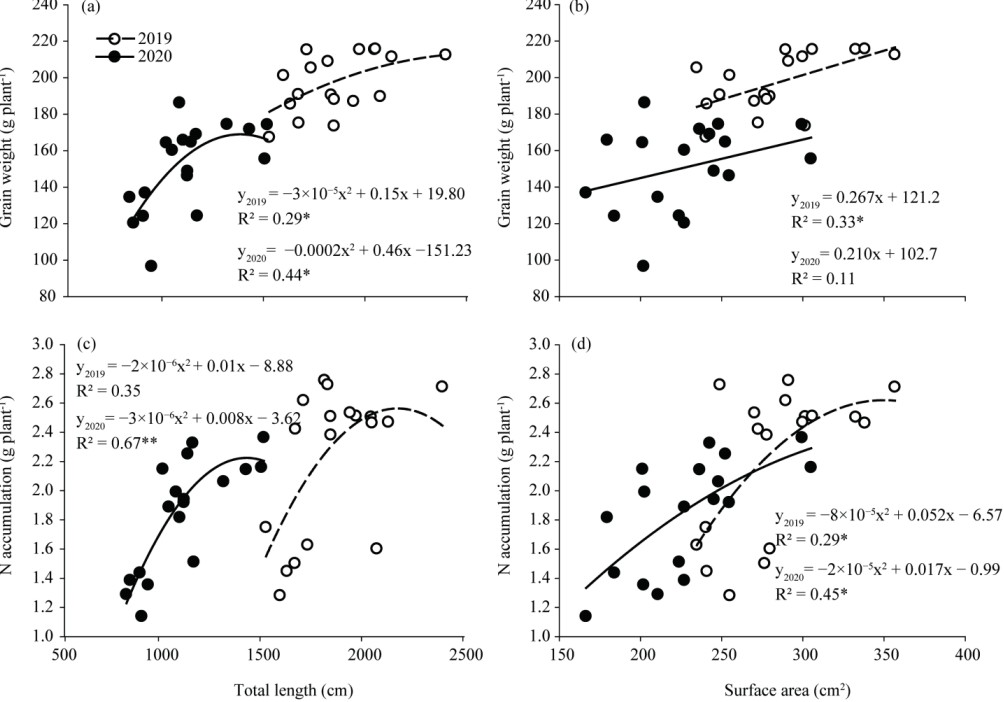

**Figure 7.** Relationship among total root length, surface area, grain yield, and N accumulation in six maize cultivars under three different N treatments. (**a**) Relationship between total root length and grain yield, (**b**) Relationship between root surface area and grain yield, (**c**) Relationship between total root length and N accumulation, (**d**) Relationship between root surface area and N accumulation. * significant at $p < 0.05$, ** significant at $p < 0.01$.

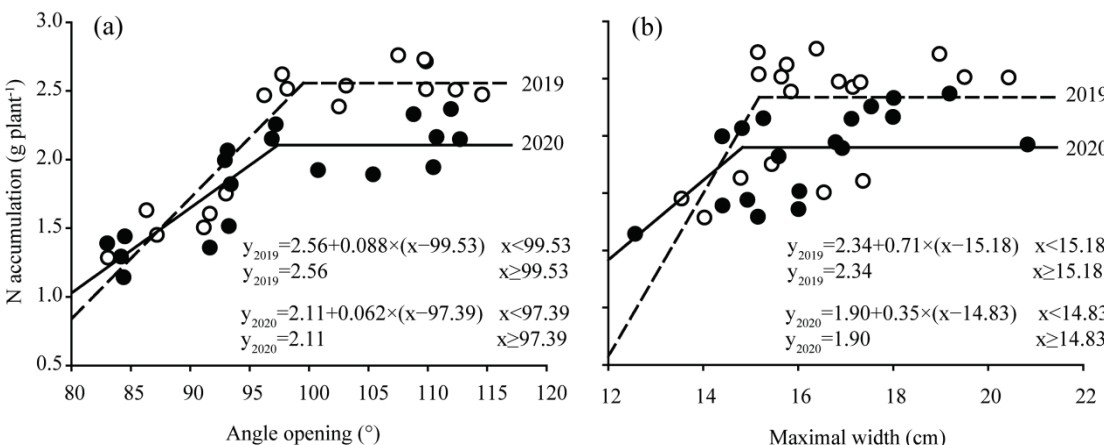

**Figure 8.** Relationship among angle opening, maximal width, and N accumulation in six maize cultivars under three different N treatments. (**a**) Relationship between root angle opening and N accumulation, (**b**) Relationship between root maximal width and N accumulation. The white and black cicles represent the 2019 and 2020 values, respectively.

## 4. Discussion

### 4.1. Influence of RSA Traits on Grain Yield and N Accumulation

Under field conditions, 80–90% of maize roots are distributed in 0–30 cm soil layers [24]. The root system of maize continues to grow and develop during the growth period, reaching the maximum at the silking stage and then begins to senesce [25]. Therefore, the silking stage is a critical period for analyzing root traits in fields [6]. The growth and development of roots are affected by genetics and the environment [26]. In this study, ANOVA showed significant effects of N levels, cultivar and years on grain yield, N accumulation and RSA (Tables 2–4). Their interactions (N × cultivar) suggested the existence of strong environmental effects on grain yield, NUE and RSA of the different maize cultivars. The differences in root biomass, root total length, surface area, root angle, and width among maize cultivars were significant. These root differences are the key factors that cause differences in N uptake and grain yield among these cultivars (Figures 6 and 7).

The function of the root system depends on the biomass and spatial distribution of the root system. Larger root systems are often closely related to higher yield, biomass and N uptake [27,28]. In this study, ZH505 and RY1210 have a larger root system and higher yield and N accumulation (Tables 2 and 3). The root length of RY1210 is longer, and it has a higher N uptake during the silking period because the root length of maize is directly related to the uptake and utilization of nitrate-nitrogen in the soil [29]. The root surface area of ZH505 is larger, and it increases the ability to obtain N in the soil through the root surface by diffusion in the rhizosphere [30]. The root system of QY9 is small, its yield and N uptake are low because the smaller root system will restrict the aboveground access to water and nutrients and will also lead to a decline in yield [5]. The angle between the maize root system and the ground is 10–80°, and the angle of the root system affects the depth of root penetration [5]. The root expansion width can reflect the horizontal distribution of the root system, ultimately affecting nutrient uptake, and gradually decreasing as the root system extends downward. In this study, RY1210 had a larger root angle and width, which can expand the root growth space and promote the uptake of nutrients and water, while the root angle of ZH505 is smaller, but the N uptake is higher, which may be because the deeper root can get more N in deep soil [6]. Although CD30 had a larger root angle and width, smaller root length and root surface area affected N absorption, while modern varieties need to have higher N acquisition capacity in deep soil [18].

Adverse weather conditions, such as droughts, elevated temperatures can reduce maize yields. In this study, the grain weight of RY1210 between 2019 and 2020 was similar at the N300 level. RY1210 has larger root width and root opening angle, which

may be important for yield in arid environments (Figure 5). Drought stress inhibited the growth of shoots and reduced the number of lateral roots, but the growth of roots usually continued [3]. The deeper the root angle, the greater the root growth angle has better water and N absorption capacity [31]. Under high planting density, competition between N and water among the plants, medium root systems with more root distribution are more likely to result in high yield [32]. RY1210 and ZH505 have the same RMW, but RO is smaller, and RA is larger in ZH505, making it more suitable for high-density planting (Table 4).

### 4.2. Correlation between RSA, Grain Yield and N Accumulation

N fertilizer application affects maize yield and N uptake by affecting root growth and distribution in the soil [33] (Tables 2–4). The growth of roots is usually improved by increasing N application, but excessive N application can inhibit the growth of lateral roots and the elongation of roots, while N deficiency can promote the increase in root biomass [16,34]. In this study, root architecture significantly correlated with yield and N accumulation, while there was no significant correlation between root angle and root width and yield (Figure 6). It may be because, under excessive N, the yield has nothing to do with root angle and root width. Studies have shown that on the clay soil of northeast China, the total root length of maize continues to increase with the increase of N application, and the total root length reaches the maximum at 168–240 kg N ha$^{-1}$, then decreases with the increase of N application [10]. In this study, excessive N application not only led to a decrease in yield but also reduced the root width (Figure 5, Table 4). Under low N conditions, the root angle and width are significantly reduced (Figure 5). This is because N deficiency forces the root system to obtain N fertilizer in deep soil, and the growth becomes steeper [35].

A robust root system is an essential feature of maize N-efficient varieties. Studies have shown that N efficient varieties have higher root length and root biomass under low and medium N conditions [3]. In this study, QY9 and CD189 had significantly reduced root weight under N deficiency, while CD30, ZH505 and RY1210 had higher root biomass (Table 3). We found that applying an appropriate amount of N fertilizer (150 kg N ha$^{-1}$) can significantly increase root angle by 19.93%, root width by 16.79%, root length by 20.39%, and root surface area by 18.58% compared with no fertilizer treatment. Therefore, the rational application of N fertilizer can promote the development of the maize root system and improve the absorption of nutrients in the soil, resulting in an efficient N absorption and high crop yield. The study found that biochar application can increase root angle by about 14%, root width about 20%, and root surface area about 54%, thereby increasing yield by 45%. The linear + plateau model revealed that when the RO reaches 97.39°, and the RMW reaches 15.18 cm, the N uptake will reach a plateau (Figure 8). However, an excessively large angle will affect the depth of root penetration, which is not efficient in the uptake and utilization of nutrients and water from deep soil and reduces the plant's ability to resist adverse environmental conditions [3]. Under high-density conditions, horizontally distributed roots reduce competition among plant roots and improve yield than vertically distributed roots [15]. Therefore, it is more suitable to plant maize varieties with medium root size at high density, because too large a root system will lead to competition between root systems, resulting in decreased yield [6].

### 5. Conclusions

Root architecture can be an important index to evaluate high-yielding and N efficient cultivars. In this study, the RSA of RL and RS displayed a significant positive correlation with grain yield and N accumulation. The RSA of RO and RMW also showed a significant positive correlation with N accumulation, no grain yield. Oversize RO and RMW cannot further improve the N accumulation. Therefore, although CD30 has larger RO and RMW, smaller RL and RS result in lower yield and N accumulation. The root system of the cultivar ZH505 and RY1210 are moderate in size, reducing the excessive carbon consumption and penetrating down to absorb N in deeper soil. The root architecture of RY1210 has good

adaptability to the arid climate. In comparison, ZH505 is suitable for high-density planting. It can be expected, selecting maize cultivars with an ideal root system architecture and applying the appropriate N fertilizer inputs in the hilly area of Southwest China can improve N efficiency and crop yield in a sustainable way.

**Author Contributions:** S.G., X.Z. and Y.Q. conceived of and designed the study; S.G., Z.L., M.H. and Z.Z. analyzed the data; S.G. wrote the manuscript; S.G., K.C., H.Y., Y.S. and Y.D. carried out the field measurements and soil analysis; F.C. and Y.L. assisted with manuscript writing and editing; T.L. and S.C. provided the material. All authors have read and agreed to the published version of the manuscript.

**Funding:** This work was supported by the National Natural Science Foundation of China (41907080), the National Key R&D Program of China (2018YFD0200700), the Sichuan Science and Technology Program (2020YFH0171), and the Research Foundation of Sichuan Academy of Agricultural Sciences (2019QYXK022, 2020BJRC005).

**Institutional Review Board Statement:** Not applicable.

**Informed Consent Statement:** Not applicable.

**Data Availability Statement:** Data can be achieved by contacting with the first author and respondense author.

**Acknowledgments:** We sincerely thank the reviewers for their helpful comments and supplementary proposal. We thank Yuanzhong Bianji for editing the English text of a draft of this manuscript.

**Conflicts of Interest:** The authors declare no conflict of interest.

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
