# Peer review of "Root System Architecture Differences of Maize Cultivars Affect Yield and Nitrogen Accumulation in Southwest China"

_agriculture, doi:10.3390/agriculture12020209_

Round 1

Reviewer 1 Report

Paper contributes new knowledge .

However, the description of tables 2, 3 and 4 needs to be expanded now is too short.

Conclusions also require elaboration, they are too laconic.

Author Response

(1) However, the description of tables 2, 3 and 4 needs to be expanded now is too short.

R1: Thanks for your suggestion. In the revised abstract at L163-168, L174-181 and L194-206, we added the description of tables 2, 3 and 4.

(2) Conclusions also require elaboration, they are too laconic.

R2: Thanks a lot for your comments. In the revised abstract at Line 377-389, we do the elaboration of conclusion.

Reviewer 2 Report

1- Abstract is to weak, it needs much improvement starting with the issue, the hypothesis, novelty and quantified results, and main implication

2- The first paragraph of the Introduction requires further information with new literature about the importance of Nitrogen on sustainable agriculture. For help, you can use the following references: (https://doi.org/10.3390/land10121375 ; https://doi.org/10.1038/nature15743 

3- The main aim and objectives of the paper not clear!!

4- Figure 1, add on the caption if this data averaged over two seasons or what?

5- Figure 5 (correlation) is difficult to read, covert it to annotated correlation matrix

6- Add future perspectives on the conclusion. 

Author Response

(1) Abstract is to weak, it needs much improvement starting with the issue, the hypothesis, novelty and quantified results, and main implication

R1: Thanks a lot for your comments. In the revised abstract at Line 27-50, we added several sentences to describe he issue, the hypothesis, novelty and quantified results, and main implication in this work.

(2) The first paragraph of the Introduction requires further information with new literature about the importance of Nitrogen on sustainable agriculture. For help, you can use the following references: (https://doi.org/10.3390/land10121375 ; https://doi.org/10.1038/nature15743 

R2: Thanks for your suggestion. In the revised mannuscript of first paragraph of the Introduction, further information with new literature about the importance of Nitrogen on sustainable agriculture were added.

(3) The main aim and objectives of the paper not clear!!

R3: In this study, our main aim was to study the genotypic differences and N response of RSA among maize varieties grown in Southwest China. We discussed the RSA to achieve high maize yield together with the efficient use of N on sustainable agriculture.

(4) Figure 1, add on the caption if this data averaged over two seasons or what?

R4: We add on the caption in Figure 1"Temperature and precipitation of Zhongjiang County during the study period in 2019 and 2020"

(5) Figure 5 (correlation) is difficult to read, covert it to annotated correlation matrix

R5: Thank you for the excellent suggestion. We covert the Table 5 to annotated correlation matrix in Figure 6.

(6) Add future perspectives on the conclusion. 

R6: Thanks for your comments. In the revised version at Line 386-389, we added future perspectives on the conclusion.

Reviewer 3 Report

Review of Agriculture 1549516; Differences in RSA among Field Grown Maize Cultivars…

The manuscript describes recent field of evaluation regionally significant maize varieties and a wide assortment of their response variables under varying N fertilization regimes. The methodology is fairly well detailed, but the data presentation is overtly systematic….almost as if devised by a template. Some readers may appreciate this methodical organization. I’ll provide advice on improvement.

L.2-4 The title is longer than recommended. Try to contain to 15 words.

L.30 Please revise ‘contributes’ to ‘contribute.’ The paper has numerous subject-verb disagreements.

L.34 Use of ‘optimal’ here seems subjective. Why not ‘Under the 150 kg N ha–1 annual regime....’ ?

L.49 & 53 Revise ‘im-portant’ to ‘important.’

L.56-65 This isn’t particularly meaningful. Maize is an annual plant. Apart from grain yield, neither society nor the authors have justified any additional appreciation for RSA.

L.98 Why monthly mean maximum temperature? Monthly mean temperature is more useful.

L.112 Revise ‘form’ to ‘from.’

L.134-135 This says the data were analyzed as a 3-way. Yet, L.83-86 indicate this is a split plot in RCBD with 3 blocks and N level as the main-plot effect. If the experiment was laid out as a split plot in RCBD, it must be analyzed as a split plot in RCBD. Meaning N level; i.e., the main-plot effect, must be F-tested by the N level x BLK term (not the residual term).

L.135-136 Having read the results, inference into cultivars and N rates are similarly thorough. So why would the authors think cultivar is being treated as random? Omit this statement. It is clear the authors are treating cultivar level (and N level) as a fixed effect.

L.151-156 I’m presuming grain yield is grain weight GW, which wasn’t statistically influenced by a N x C x Y interaction. Table 2 then describes the main effects of N or C on GW adequately. Figure 3 is superfluous and should be omitted.

L.156-161 The methods should explain how N accumulation (kg hm–2) is calculated. The unit hm–2 isn’t an SI unit. Revise to m–2 or ha–1.

L.157-165 The calculation of N accumulation is not described. It does not appear to have been analyzed as a dependent variable either. There is no evidence that it is statistically influenced by a N x C x Y interaction. Thus, Figure 4 is superfluous and should be omitted.

Relative to 2019, the vast majority of dependent variables measured statistically decreased in 2020. The 2020 season also appeared much drier than 2019 in Apr., May, and June. In L.66-68, the authors  indicate drought and poor soil fertility are regional challenges and seek to identify maize cultivars with ideal RSA suitable for this environment. Was there a specific N level where any of these 6 cultivars produced 2020 yields that were not significantly less than those recorded in 2019? If not, what difference did RSA really make?

The paper wasn't ready for review. The authors describe N uptake or N accumulation, the aforementioned units as g plant-1 or kg hm–2, interchangeably. Same with grain weight and grain yield. It is distracting. If two y-axes are presented side-by-side, and the right-hand axis doesn’t indicate units, the reader assumes the left axis units applies to both. Yet, this wasn’t the case in Figure 8. L.247-251 Figure 8b and 8d do not have units on their y-axes.

If the authors are going to ignore the effect of year in the results and discussion, they should re-analyze specifying year as random. They can then display all the significant C x N interactions in a more simple fashion. In future studies, when the experimental objective is to identify N fertilization rates for maximum yield, the authors should incorporate more than two non-zero N levels.

Author Response

(1) L.2-4 The title is longer than recommended. Try to contain to 15 words.

R1: Yes, it has been modified as "Root System Architecture Differences of Maize Cultivars Affect Yield and Nitrogen accumulation in Southwest China"

(2) L.30 Please revise ‘contributes’ to ‘contribute.’ The paper has numerous subject-verb disagreements.

R2: Yes, it has been modified

(3) L.34 Use of ‘optimal’ here seems subjective. Why not ‘Under the 150 kg N ha–1 annual regime....’ ?

R3: Yes, it has been modified

(4) L.49 & 53 Revise ‘im-portant’ to ‘important.’

R4: Yes, it has been modified

(5) L.56-65 This isn’t particularly meaningful. Maize is an annual plant. Apart from grain yield, neither society nor the authors have justified any additional appreciation for RSA.

R5:Thanks for your suggestion. We have added RSA and yield research advances at L.69-81.

(6) L.98 Why monthly mean maximum temperature? Monthly mean temperature is more useful.

R6:Because the website gives the monthly mean maximum temperature. We find another website with monthly mean temperature (https://stjj.suining.gov.cn) and modify Figure 1.

(7) L.112 Revise ‘form’ to ‘from.’

R7:Yes, it has been modified

(8) L.134-135 This says the data were analyzed as a 3-way. Yet, L.83-86 indicate this is a split plot in RCBD with 3 blocks and N level as the main-plot effect. If the experiment was laid out as a split plot in RCBD, it must be analyzed as a split plot in RCBD. Meaning N level; i.e., the main-plot effect, must be F-tested by the N level x BLK term (not the residual term).

R8: Thank you very much for this comment. In this study, our experiments were run in the long-term nitrogen fertilizer application fields. Three N treatments: N0, N150, and N300 has been applied since 2016. In previous study, no statistical differences were observed between N level and blocks[1]. Therefore, we ignored the effect of block in this study.

[1]Guo,S.; Zeng, X.Z.; Chen, K.;Lu, T.Q.; Shang G.Y.X.; Zhou, Z.J.; Tu, S.H.;Qin, Y.S.Screening of N-efficient maize hybrids and analysis for their yield increase potentials in central Sichuan Basin. Journal of Nuclear Agricultural Sciences2020, 34,2567-2577.

(9) L.135-136 Having read the results, inference into cultivars and N rates are similarly thorough. So why would the authors think cultivar is being treated as random? Omit this statement. It is clear the authors are treating cultivar level (and N level) as a fixed effect.

R9: Thank you very much for this comment. You are right. We rephrased this sentence in the revised version at Line 155-156. N level and cultivar were treated as fixed effect.

(10) L.151-156 I’m presuming grain yield is grain weight GW, which wasn’t statistically influenced by a N x C x Y interaction. Table 2 then describes the main effects of N or C on GW adequately. Figure 3 is superfluous and should be omitted.

R10:Thank you for your suggestion. We take Figures 3 be omitted.

(11) L.156-161 The methods should explain how N accumulation (kg hm–2) is calculated. The unit hm–2 isn’t an SI unit.

R11:Thanks for your good advice. We revise to m–2 or ha–1. N accumulation per unit area (kg ha-1) = whole-plant N content at maturity × density per unit area.

(12) L.157-165 The calculation of N accumulation is not described. It does not appear to have been analyzed as a dependent variable either. There is no evidence that it is statistically influenced by a N x C x Y interaction. Thus, Figure 4 is superfluous and should be omitted.

(13) R12:Thank you for your suggestion. We take Figures 4 be omitted.

Relative to 2019, the vast majority of dependent variables measured statistically decreased in 2020. The 2020 season also appeared much drier than 2019 in Apr., May, and June. In L.66-68, the authors  indicate drought and poor soil fertility are regional challenges and seek to identify maize cultivars with ideal RSA suitable for this environment. Was there a specific N level where any of these 6 cultivars produced 2020 yields that were not significantly less than those recorded in 2019? If not, what difference did RSA really make?

R13: Yes, We found that there was no significant difference in the grain weight of RY1210 between 2019 and 2020 by T-test at N300 level. RY1210 is used for larger root width and root opening angle, which may be important RSA to ensure yield in arid environments(Figure 5). We added a discussion at L330-340..

(14) The paper wasn't ready for review. The authors describe N uptake or N accumulation, the aforementioned units as g plant-1or kg hm–2, interchangeably. Same with grain weight and grain yield. It is distracting. If two y-axes are presented side-by-side, and the right-hand axis doesn’t indicate units, the reader assumes the left axis units applies to both. Yet, this wasn’t the case in Figure 8. L.247-251 Figure 8b and 8d do not have units on their y-axes.

R14:We supplement y-axis units in Figure 8. The unit of nitrogen accumulation is unified as g plant-1.

(15) If the authors are going to ignore the effect of year in the results and discussion, they should re-analyze specifying year as random. They can then display all the significant C x N interactions in a more simple fashion. In future studies, when the experimental objective is to identify N fertilization rates for maximum yield, the authors should incorporate more than two non-zero N levels.

R15: Thank you very much for this comment. In this study, our ANOVA results showed years have significant effects of grain yield, N accumulation and root architecture traits. Therefore, root architecture traits and the relationship among root traits and grain yield, N accumulation were displayed in 2019 and 2020, separately. In addition, thank you for your suggestion about incorporate more than two mom-zero N levels to identify N fertilization rates for maximum yield. You are absolutely right, in the future studies, we plan to add N180 and N250 two more N application rates to the optimize linear + plateau model (Figure 8) to evaluate the relationship between grain yield and N fertilizer application rates.

Round 2

Reviewer 3 Report

OK, thank you.